# Increasing Fat Deposition Via Upregulates the Transcription of Peroxisome Proliferator-Activated Receptor Gamma in Native Crossbred Chickens

**DOI:** 10.3390/ani11010090

**Published:** 2021-01-05

**Authors:** Supanon Tunim, Yupin Phasuk, Samuel E. Aggrey, Monchai Duangjinda

**Affiliations:** 1Department of Animal Science, Faculty of Agriculture, Khon Kaen University, Khon Kaen 40002, Thailand; non_300928@hotmail.com (S.T.); yuplua@kku.ac.th (Y.P.); 2Research and Development Network Center for Animal Breeding (NCAB), Faculty of Agriculture, Khon Kaen University, Khon Kaen 40002, Thailand; 3NutriGenomics Laboratory, Department of Poultry Science, University of Georgia, Athens, GA 30602, USA; saggrey@uga.edu

**Keywords:** PPAR, gene expressions, fat deposition, Thai native crossbred chickens

## Abstract

**Simple Summary:**

Crossbreeding using exotic breeds is usually employed to improve the growth characteristics of indigenous chickens. This mating not only provides growth but adversely affects excess fat deposition as well. This deposition was regulated by a complicated cellular mechanism including peroxisome proliferator-activated receptors (PPARs) function. Thus, we hypothesized that native chickens breed percentage might be related to PPARs gene expression. This study aimed to study the role of PPARs on fat deposition in chickens which was the different native genetic background. Our results indicated that increasing commercial breed percentage in the chicken leads to increased fat deposition via the increasing of PPARG gene expression. Therefore, the PPARG gene notable as a major gene of cellular fat deposition and might be applied in further study.

**Abstract:**

This study aimed to study the role of PPARs on fat deposition in native crossbred chicken. We studied the growth, abdominal, subcutaneous, and intramuscular fat, and mRNA expression of PPARA and PPARG in adipose and muscle tissues of four chicken breeds (CH breed (100% Thai native chicken), KM1 (50% CH background), KM2 (25% CH background), and broiler (BR)). The result shows that the BR chickens had higher abdominal fat than other breeds (*p* < 0.05) and the KM2 had an abdominal fat percentage higher than KM1 and CH respectively (*p* < 0.05). The intramuscular fat of BR was greater than KM1 and CH (*p* < 0.05). In adipose tissue, PPARA expression was different among the chicken breeds. However, there were breed differences in PPARG expression. Study of abdominal fat PPARG expression showed the BR breed, KM1, and KM2 breed significantly greater (*p* < 0.05) than CH. In 8 to 12 weeks of age, the PPARG expression of the CH breed is less than (*p* < 0.05) KM2. Crossbreeding improved the growth of the Thai native breed, there was also a corresponding increase in carcass fatness. However, there appears to be a relationship between PPARG expression and fat deposition traits. therefore, PPARG activity hypothesized to plays a key role in lipid accumulation by up-regulation.

## 1. Introduction

In tropical climate countries, native chickens selectively bred from jungle fowl are distributed widely. In Asian countries, they are numerous slow-growing and lean breeds [1]. These indigenous poultry strains generally have low fat and the ability to tolerate the effect of heat stress. On the other hand, commercial broilers are selected for rapid growth and do not express their full genetic potential under hot-humid climates.

In tropical regions, crossbreeding of indigenous breeds with improved exotic breeds has been one of the strategies to improve the growth of the indigenous breeds. Crossbreeding has been used in many countries including Nigeria, Thailand, Bangladesh, and India to improve growth and egg production [2]. High producing exotic commercial broiler strains have high growth performance and a significant amount of visceral fat compared to the indigenous breeds. Crossbreeding can also increase fat deposition in both intramuscular fat and subsequently affect meat flavor [3]. Excessive carcass fatness can also have a negative effect on the dressing percentage of chicken and consumer health due to the high-fat content of the meat.

It is important to understand the lipogenesis mechanism in poultry crossbreeds to ascertain the optimal level of infusion of exotic (genes) alleles into indigenous breeds to take advantage of the growth potential and simultaneously limiting visceral fat. The metabolism of avian species is complex and may have a different mechanism from mammalian species [4,5]. Lipogenesis is mediated in part by fatty acid-binding proteins (FABPs), lipoprotein lipase (LPL) and fatty acid synthase (FAS), and the transcriptional expression of nuclear receptors [6].

The peroxisome proliferator-activated receptors (PPARs) are a superfamily of nuclear receptors that play a significant role in adipocyte cell differentiation and intra- and extracellular transportation of fatty acid [5,7]. PPARG is one of the most important subtypes, which has activity in oil droplet accumulation by regulated glucose and fatty acid uptake and directly combines with PPAR response element (PPRE) [8]. Contrary to PPARG, PPARA serves in lipid catabolism, especially β-oxidation via upregulating related enzymes [9].

The objective of the current study was to investigate the mRNA expression of nuclear receptors, PPARA and PPARG in chicken adipose and muscular tissues in Thai native chickens with different percentages of commercial broiler genotypes.

## 2. Materials and Methods

### 2.1. Birds and Rearing Condition

This study was approved by the Institute of Animal for Scientific Purpose Development (IAD, IACUC-KKU-34/62). We received purebred and crossbred Thai native chickens from the Research and Development Network Center for Animal Breeding of Khon Kaen University. The Arbor Acre commercial broiler used in this study was purchased as day-old chicks from Charoen Pokphand Company. The study was conducted with four genotypes. The native Thai breed, Chee (CH) (100% Thai native chicken background: 0% broiler background), CH male and broiler female (Kaimook e-san1; KM1) (50% Thai native chicken background: 50% broiler background), broiler male and KM1 female (Kaimook e-san2; KM2) (25% Thai native chicken background: 75% broiler background), and broiler (BR) (0% Thai native chicken background: 100% broiler background). Chickens were raised under the simulated condition of Thai native and its crossbred production with the same management and husbandry conditions and fed with a commercial broiler diet throughout the experiment for a response to the maximum genetic potential of all studied breeds. The poultry house was an open-air system. The poultry house was an open-air system. There were four pens per genotype and 25 birds per pen. All chickens were fed ad-libitum on a diet consisting of 21% crude protein (CP), 3100 kcal of ME/kg for starter diet, and 19% CP and 3200 kcal of ME/kg for growing diet.

### 2.2. Slaughtering, Fat Deposition Data, and Tissue Collection

Twenty birds (10 males and 10 females) were randomly selected per genotype at 6, 8, 10, and 12 weeks of age for CH, KM1, and KM2. The commercial broiler was slaughtered only at 6 weeks of age because of the limitations of raising broilers in open systems.

After the slaughtering process, blood and feathers were discarded from the carcass. Abdominal fat, including around the gizzard and proventriculus, was immediately collected and weighed. Whole-body skin with subcutaneous fat, except in wingtip were carefully isolated and weighed and designated as abdominal- and subcutaneous fat percentage expressed as a percentage of carcass weight. Sliced right breast (Pectoralis major), right thigh (Bicep femoris), and skin (whole breast skin) of chickens (*n* = 8/per week per breed) were collected into a vacuum bag and stored at −20 °C until further analysis. The samples were analyzed for the fat percentage determination following the AOAC method [10] by dried samples in triplicates were extracted with petroleum ether in Soxhlet extraction. Samples of P. major and abdominal fat (*n* = 8) per week per genotype were taken from the central portion of these tissues into an insulated bag and snap-frozen by liquid nitrogen and stored at −20 °C freezer until used in RNA extraction.

### 2.3. Quantitative Reverse Transcription Polymerase Chain Reaction (qRT-PCR)

Total RNA was extracted from randomized parts of tissue within P. major and abdominal fat tissues using GeneJET RNA Purification Kit (Thermo Scientific, Waltham, MA, USA). Concentrated and purified RNA yield was then quantified by The NanoDrop™ 2000/2000c Spectrophotometer (Thermo Scientific, USA). Ratios of absorption (260/280 nm) of all extracted RNA were in the range of 1.8–2.0. Qualified RNA products were stored at −20 °C until use in further analysis.

Bio-Rad CFX96 Touch Real-Time PCR Systems (Bio-Rad, Hercules, CA, USA), with optical grade plates using IQTM PCR plate (Bio-Rad, USA) was used in quantitative reverse transcription-polymerase chain reaction (qRT-PCR) analyses. We analyzed the RNA expression of PPARA and PPARG using 18s rRNA as a reference gene for normalization. Sequencing, fragment size, and annealing temperature of the primers are presented in Table 1. [11,12]. Primers were purchased from 1st BASE Oligonucleotide Synthesis (1st Base, Singapore). The single-step RT-PCR was used to investigate target gene expression by the SensiFAST™ SYBR^®^ No-ROX One-Step Kit (Bioline, Memphis, TN, USA). The sample was amplified in duplicate. Each bird was used as an experimental unit which used the 2^−△△Ct^ method to assess the fold change [13] using the CH genotype as the control.

### 2.4. Statistical Analysis

All data including fat deposition and the gene expression were evaluated assumption of ANOVA by PROC univariate procedure before analyzed by ANOVA using the Generalized Linear Model (GLM) procedure by SAS statistical software package, version 9.0 (SAS Institute, Inc., Cary, NC, USA), and the means separated by Tukey. The phenotypic correlation was calculated between gene expression and fat deposition traits using Pearson correlation.

## 3. Results

### 3.1. Carcass Fat Deposition among Various Genetic Background of Chickens

The fat deposition in the abdominal, subcutaneous tissue, skin, P. major, and thigh muscle were investigated in different breeds (Figure 1). The CH breed had the lowest fat deposition (*p* < 0.05) compared with KM1, KM1, and BR for all the tissues studied. At week 6, the BR breed had a significantly higher fat percentage in all tissues except skin fat compared to the other breeds. From 8 to 12 weeks, the comparison was between CH, KM1, and KM2. The KM2 breed had the highest percentage (*p* < 0.05) of abdominal fat compared to KM1 and CH). Skin percentage and skin fat in both KM1 and KM2 were not different (*p* > 0.05). There is a significantly higher intramuscular fat (IMF) of BR in both P. major and B. femoris than KM1 and CH (*p* < 0.05) but not different with KM2. From 8 to 12 weeks, CH breed was lower IMF than KM2 in both muscle tissues (*p* < 0.05). While there was no difference in intramuscular fat between KM1 and KM2 chickens.

### 3.2. Differentiation of PPARA mRNA Expression in Various Breed of Chickens

Transcriptional levels of PPARA in abdominal fat and breast muscular tissue are demonstrated in Figure 2I and Figure 3I respectively. In this current study, we found a significant difference in PPARα expression between BR and CH (*p* < 0.05) at 6 weeks of age. While no difference could be found among expression by KM1, KM2, and BR. At 8 weeks of age, CH breed expressed PPARA mRNA levels less than (*p* < 0.05) KM1 breed which was not different between the expression of both crossbred chickens. However, during 10 to 12 weeks of age, there are no differences in PPARA expression between breeds in abdominal fat tissue. There was no difference in muscular PPARA expression levels among studied breeds throughout the study.

### 3.3. Differentiation of PPARG mRNA Expression in Various Breed of Chickens

Abdominal fat and breast muscular tissue were investigated for PPARG gene expression level for breed comparison as shown in Figure 2II and Figure 3II. Abdominal fat PPARG gene expression in BR, KM1, and KM2 breeds had higher mRNA expression levels (*p* < 0.05) than CH breed but not found different expression level among BR, KM1, and KM2 at 6 weeks of age. At the 8th week of age, CH and KM1 had similar mRNA expression but both had lower PPARG expression (*p* < 0.05) when compared with KM2. At 12 weeks of chicken age, the results show that the PPARG expression of the CH breed was less than (*p* < 0.05) in both crossbred breeds of chicken.

In the P. major, PPARG expression was higher (*p* < 0.05) in BR compared to the other breeds at 6 weeks of age, however, there were no differences among CH, KM1, and KM2. At 8 and 10 weeks of age, PPARG expression was higher in KM2 compared to KM1 and CH and the differences between CH were not significant.

### 3.4. Correlation between PPARs Expression and Fat Deposition in Slaughtering Trait

Pearson correlation coefficient was calculated from all breeds 6 to 12 weeks of age for PPAR expressions and fat deposition traits (Table 2). PPARG transcriptional level in abdominal fat tissue showed a positive correlation with adipose tissue accumulation in the abdomen and skin structure (rp = 0.33–0.38). PPARA inversely correlated with skin fat at −0.26. Intramuscular fat in both breast and thigh tissues had a moderate positive correlation with PPARG expression (0.34–0.43). On the other hand, PPARA did not correlate with fat depositions in both adipose and muscular tissues. There was a positive correlation between muscular PPARG expression and adipose tissue fat accumulation which were 0.37, 0.26, and 0.20 for abdominal, subcutaneous, and skin fat, respectively.

## 4. Discussion

### 4.1. Impact of Crossbreeding of Thai Native Chickens on Growth

Indigenous chickens are important in developing countries for food security and the socio-cultural life of the rural community [2]. The Thai native chickens are an important genetic resource, as they are adapted to the harsh environmental conditions and have a chewy texture and taste that are preferred by consumers in Thailand [1]. Despite these advantages, the growth performance of the native chickens is poor and as a result, crossbreeding with fast-growing exotic strains has been encouraged to improve the growth characteristics of the native breeds. In the current study, the KM1 and KM2 crossbred chickens with 50 and 25%, respectively of native Thai chicken background showed significant improvement in growth. The KM1 and KM2 chickens grew about 1.3 and 2.3 folds, respectively when compared with the CH breed. The growth performance of these studied breeds was handled by an adequate nutrient in the commercial broiler diet to meet a nutrient requirement and respond to the genetic potential of growth and fat deposition as well.

The improvement in growth of the native Thai breed has come with an associated increased in body fatness. Abdominal fat, which is an excessive fat and considered as waste in the slaughtering process. There is a clear positive relationship between growth performance and abdominal fat accumulation. The fat deposition has been correlated with adipocyte enlargement [11]. Improvement in growth and accumulation of fatness associated with crossing breeding of indigenous chickens with exotic breeds have been documented in several other native crossbreeding programs [14]. This has been shown to be due to the pleiotropic effect between body weight and abdominal fat traits [15]. With the faster growth rate of the KM2 chickens because of the higher percentage of exotic genes, it is expected that they will reach slaughter age much faster than the KM1 crossbred and the CH breed.

Contrarily to expectation, the KM1 and KM2 did not show any differences in subcutaneous and skin fat, but they both had values higher than the CH breed. It has been documented that; the weight of the skin is dependent on the amount of subcutaneous fat deposition [15]. However, from the current study, the phenotypic correlation between skin and subcutaneous fats was 0.54. There may be other non-genetic factors contributing to the relationship between subcutaneous fat and skin fat.

The intramuscular fat (IMF) represents the lipid that is distributed in muscular tissue containing epimysium, perimysium, and endomysium which infiltrates between the muscular fiber bundles. IMF has an influence on meat quality which varies depending on sex, slaughter age, and type of muscle [16]. The CH breed has relatively low IMF accumulation in both breast and leg muscle while BR and KM2 have a significantly high amount of IMF. The current report suggests that crossbreeding of the Thai native chickens with exotic breeds have the potential to change not only the growth performance but the meat quality as well.

Zhou et al. [17] demonstrated that the meat from selected for increased fat content have a lower shear force than their control counterparts. Potentially, the meat of the KM1 and KM2 may be more tender than the native CH breed due to their relatively high IMF. However, [18] did not observe any differences in IMF between native Thai and Barred Plymouth Rock crossbred and the native chickens when breast and thigh muscles were compared. Similarly, crossbreeding Chinese native chickens did not affect breast IMF [14].

### 4.2. PPARs Transcription Factors Regulating Cellular Lipid Metabolism

In chickens, endogenous lipids are mainly synthesized as lipoproteins in the liver from dietary glucose and then export to extrahepatic tissues by circulation in the bloodstream, where the lipoproteins are hydrolyzed by lipoprotein lipase and fatty acids are released for use as energy or accumulation in the cell [16,19]. At the cellular level, the many types of functional proteins related to lipid metabolism were reviewed in [20] that included Fatty Acid Binding Proteins (FABPs), insulin-dependent glucose transporter 4 (GLUT4), lipoprotein lipase (LPL), and the fatty acid translocase (CD36), which were stimulated by transcription factor PPAR. PPARs are important cellular regulators that respond to energy status during both fed and fasted states [21]. The PPARs function and mechanism in mammalian species were studied, while many studies focused on the differences in lipid metabolism at the molecular level between avian and other species, and the unique PPARs function in chicken lipid metabolism [22].

PPARA is one of the transcription factors involved in the regulation of the ketogenesis pathway [21]. the PPARA Mitochondrial 3-hydroxy-3-methylglutaryl-CoA synthase (HMGCS2) interaction which is a nodal point in the ketogenic mechanism and this complex is transported to the nucleus where it activates PPRE to encode the transcription of HMGCS gene for autoregulation of its own nuclear transcription [23].

At 6 weeks of age, PPARA mRNA expression in adipose tissue was significantly downregulated in the commercial breed of chicken (BR) compared to CH and KM1. The downregulation may be the effect of cellular energy conservation of Brand KM2 and lead to the remaining fat to enlargement of abdominal fat tissue. From the current study, it appears that mRNA expression of PPARA is not directly dependent on the genetic background due to there was in the stage that unnecessary to use visceral fat as energy sources. In the current study, PPARA transcriptional level in the muscle (Figure 3) did not differ among the breeds studied. This may be due to the muscle type as breast muscle require low energy because of lack of movement. We did not observe any significant phenotypic correlation between PPARA mRNA expression level and IMF in both breast and thigh tissues. Thus, the increase in IMF with the muscles of the Thai native crossbreds was not due to changes in PPARA transcription. On the contrary, [24] reported a positive relationship between PPARA expression and IMF deposition in dwarf chicken which has deletion mutation in 3’UTR of GHR leading to a reduction of body weight and increased IMF accumulation.

However, there appears to be a relationship between PPARG expression and IMF. The PPARG mRNA expression of BR and crossbred chicken (KM1 and KM2) were in concordance with the fat deposition traits. PPARG is one of the most important subtypes, which has activity in oil droplet accumulation within adipose tissue. However, there are many related mechanisms such as glucose and fatty acid uptake regulation by LPL, GLUT, and A-FABP [8,25]. Therefore, these lipid accumulations of BR, KM1, and KM2 could occur putatively via cellular uptake and transport fatty acid for storage as triglycerides. Moreover, the coefficient correlation revealed that PPARG expression has a moderate positive correlation with abdominal and skin fat. Moreover, Wang et al. [26] showed that the A-FABP gene is down-regulated when PPARG is silenced, therefore, PPARG activity hypothesized to plays a crucial role in cellular lipid accumulation by A-FABP activity especially in lipogenesis, and may have potential as a target gene for selection against excessive fat deposition in chickens. We have shown that A-FABP is upregulated in concordance with the fatness level of the breed [27]. The correlation between PPARG expression and IMF for both breast and thigh muscles was positive. This is corroborated by other studies using Chinese native chickens and female Wuhau chickens [28].

PPARG was elucidated as a role factor in fat deposition in both adipose and muscular tissue that might affect by miRNA according to a previous study. [29] reported the function of miRNA-122 in adipose tissue by opposite expression trend to PPARG. Intramuscular fat deposition also reported a relationship with miRNA function showed that the expressed miRNAs and there were involved in energy metabolism, glycerophospholipid metabolism, fatty acid elongation, and degradation pathways, insulin signaling, and PPAR function via miRNA-499-5p/SOX6 and miRNAs-196-5p/CALM1 [30,31]. This adipogenic regulation was described by [30] to demonstrate that intramuscular adipocyte differentiation was controlled via miRNA-140-5p promote targeting retinoid X receptor gamma, which co-activated function with PPAR and lead to mRNA levels of the PPARG and A-FABP increased with adipocyte differentiation.

## 5. Conclusions

We studied the relationship between abdominal, subcutaneous, and intramuscular fat in Thai native crossbreds, an exotic commercial broiler, and the Thai native breed, Chee. Crossbreeding significantly improved growth with a concordance increase in carcass fatness. There are appears to be no relationship between fatness and mRNA expression of PPARA. However, the transcriptional expression level of PPARG in both adipose and muscular tissue seems to correlate with the amount of fatness in the breed studied. It is thought that the inclusion of exotic genes in the Thai native chickens may also affect the meat quality, even though carcass fatness may also increase.

## Figures and Tables

**Figure 1 animals-11-00090-f001:**
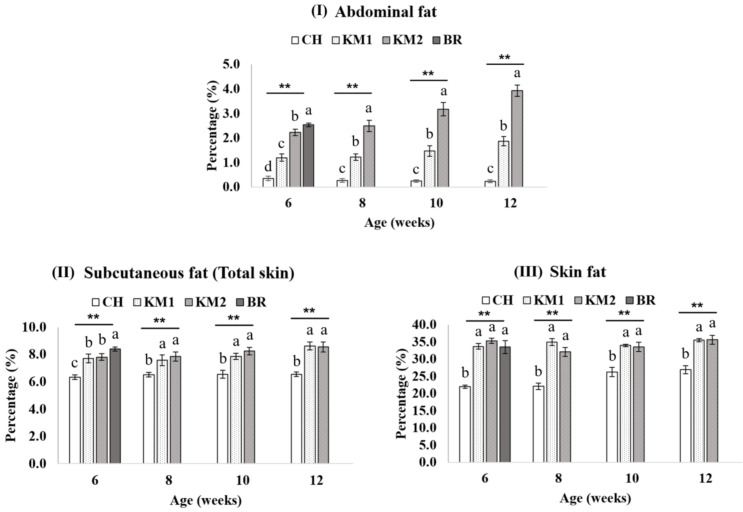
The fat deposition traits including abdominal fat (**I**), subcutaneous fat (**II**), skin fat (**III**), breast intramuscular fat (**IV**), and thigh intramuscular fat (**V**) comparison between breeds of chicken in 6–12 weeks of slaughtering age. CH = Chee, KM1 = Kaimook e-san1, KM2 = Kaimook e-san2, BR = Broiler chicken. a, b, c and d Mean values within a figure with no common letter differ significantly (* mean *p* < 0.05 and ** mean *p* < 0.01).

**Figure 2 animals-11-00090-f002:**
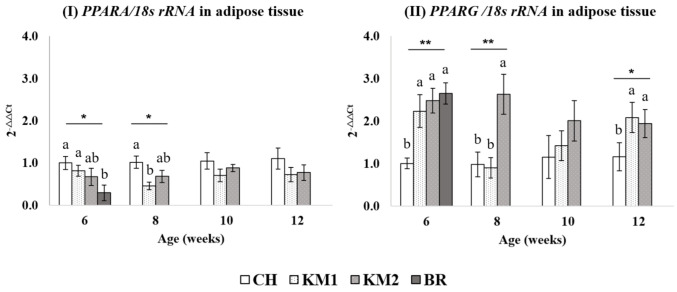
The transcriptional expression level of PPARA (**I**) and PPARG (**II**) normalized with 18S rRNA in abdominal fat tissue. a, b Mean values within a figure with no common letter differ significantly (* mean *p* < 0.05 and ** mean *p* < 0.01).

**Figure 3 animals-11-00090-f003:**
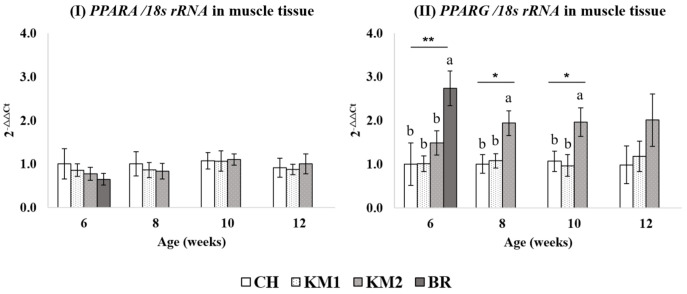
The transcriptional expression level of PPARA (**I**) and PPARG (**II**) normalized with 18S rRNA in breast muscle tissue. a, b Mean values within a figure with no common letter differ significantly (* mean *p* < 0.05 and ** mean *p* < 0.01).

**Table 1 animals-11-00090-t001:** Primer sequence, PCR product size, and annealing temperature.

Genes	Sequences	Product (bp)	TM	Sources
PPARA	F: 5- AGGCCAAGTTGAAAGCAGA-3 R: 5-GTCTTCTCTGCCATGCACAA-3	217	58	[11]
PPARG	F: 5-GACCTTAATTGTCGCATCCAT-3 R: 5-CGGGAAGGACTTTATGTATGA-3	237	56	[11]
18S rRNA	F: 5-CGGCGACGACCCATTCGAAC-3 R: 5-GAATCGAACCCTGATTCCCCGTC-3	99	62	[12]

**Table 2 animals-11-00090-t002:** Pearson correlation coefficient between PPARs mRNA expression and fat deposition traits in both adipose and muscle tissues.

Fat Deposition Traits (%)	Adipose Tissue		Muscular Tissue	
	PPARA (*n* = 102)	PPARG (*n* = 102)	PPARA (*n* = 102)	PPARG (*n* = 102)
Abdominal fat (*n* = 258)	−0.02	0.38 **	−0.02	0.37 **
Subcutaneous fat (*n* = 260)	−0.18	0.10	0.01	0.28 **
Skin fat (*n* = 104)	−0.26 *	0.32 **	−0.06	0.20 *
Breast intramuscular fat (*n* = 104)	−0.23 *	0.22 *	−0.04	0.39 **
Thigh intramuscular fat (*n* = 104)	−0.28 **	0.36 **	−0.04	0.46 **

* and ** mean significantly correlation coefficient at * *p* < 0.05 and ** *p* < 0.01 level.

## Data Availability

The data presented in this study are openly available in Research square at https://www.researchsquare.com/article/rs-32691/v1. The additional data other than that are available on request from the corresponding authors.

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
