# Peer review of "Increasing Fat Deposition Via Upregulates the Transcription of Peroxisome Proliferator-Activated Receptor Gamma in Native Crossbred Chickens"

_animals, 2021, doi:10.3390/ani11010090_

Round 1

Reviewer 1 Report

This is a very interesting manuscript which investigates hybrid chicken from indigenous breeds crossed with exotic breeds. This cross leads to more productive hybrids but the authors note that there is an inappropriate deposition of abdominal fat. Their conclusion is ‘However, there appears to be a relationship between PPARγ expression and fat deposition traits. therefore, PPARγ activity plays a key role in lipid accumulation by up-regulation. ‘

The authors confuse correlation with causation. They have no causative data in this manuscript. They can say ‘may play’ or is’ hypothesised to play’ a key role. There is no proof that it does.

Major comments

Line 106 We analyzed the RNA expression of PPARα and PPARγ using 18s rRNA as a reference gene for normalization.

Normallising low expressed genes to highly expressed genes is problematic. How have the authors taken this into account? How do the authors normalise expression in muscle tissue. Is there more fat cells in the muscle tissue? Then it would be expected that PPAR-g would increase if the authors normalise to 18S levels which is also expressed in muscle cells. Perhaps the authors could normalise PPAR g levels to PPAR a levels? If differences still exist, the results may be valid.

Line 77 Chickens were raised under the same management and husbandry conditions and fed with commercial broiler diet throughout the experiment.

This was ill thought out and should be discussed in the discussion. It is quite obvious that feeding a broiler diet would increase abdominal fat in the hybrids. It would be more interesting to measure growth of all these birds under the dietary conditions of the indigenous breed.

Line 47. Commercial breeds of Gallus gallus also evolved from red jungle fowl. Why do you draw a distinction? Perhaps use the word selectively bred instead of ‘evolved’.

Minor comments:

It is not clear in the abstract that KM1 and KM2 are cross breeds with the broiler breed. It also should be apparent that the authors are interested in meat production, not egg production. Usually, the purpose of crossbreeding programmes are to produce improved dual purpose chickens. Using a broiler chicken is a strange choice for a hybrid breeding expt.

Line 25 and throughout manuscript. Fraction is the incorrect word. Pecentage is a more appropriate word.

Line 81 The Arbor Acre commercial broiler used in this study was purchased.

You need to make it clear that you have purchased hatching eggs. These will be from a four way cross so the genetics in the hybrids will be highly varied.

Line 92 2.2. Slaughtering, fat deposition data and tissue collection

The authors do not mention that the broilers were slaughtered at 6 weeks of age.

Author Response

Revision list

Manuscript ID: animals-1028268 “Increasing Fat Deposition via Upregulates the Transcription of Peroxisome Proliferator-Activated Receptor Gamma (PPARγ) in Native Crossbred Chickens”

Dear Dr. Cristina Maria Fiat and Reviewer,

We appreciate your responsibility as an editor and assistance with this manuscript. The reviewers have provided several useful suggestions to our manuscript and we have revised to the best of our ability. Our revised manuscript has modifications in yellow highlighted text, please let me know if I have to provide additional information.

Reviewer #1

  1. Normalising low expressed genes to highly expressed genes is problematic. How have the authors taken this into account? How do the authors normalise expression in muscle tissue. Is there more fat cells in the muscle tissue? Then it would be expected that PPAR-g would increase if the authors normalise to 18S levels which is also expressed in muscle cells. Perhaps the authors could normalise PPAR g levels to PPAR a levels? If differences still exist, the results may be valid.

Response: We appreciate your valuable comments to improve the quality of the manuscript. The normalizing gene as 18s rRNA was determined following the method of Livak and Schmittgen (2001) and a previous study (Dridi et al., 2007; Wang et al., 2012) which used 18s rRNA for normalized gene expression in muscular, adipose, and other tissues. Therefore, according to your suggestion, the PPARG was normalized with PPARA in adipose and muscle tissue were reanalyzed and reported in figure 1.

Figure 1. Transcriptional expression level of PPARG normalized with PPARA in abdominal fat tissue (I) and muscular tissue (II). a, b Mean values within a figure with no common letter differ significantly (* mean p< 0.05 and ** mean p<0.01)

            According to results from PPARG normalized with PPARA the overall results have a minor change but the significant difference between PPARG relative expression still exists especially between KM2 and CH breed. Therefore, the crossbreeding with commercial breed percentage should be upregulating PPARG gene expression and hypothesized that may be related to increased fat deposition.

  1. Line 77 Chickens were raised under the same management and husbandry conditions and fed with commercial broiler diet throughout the experiment.

This was ill thought out and should be discussed in the discussion. It is quite obvious that feeding a broiler diet would increase abdominal fat in the hybrids. It would be more interesting to measure growth of all these birds under the dietary conditions of the indigenous breed.

Response: In this study, we used a commercial broiler diet because of the reason as following:

- We would like to make sure that an available nutrient in the diet was adequate for all studied breed especially broiler chicken and KM2 which has a high percentage of the commercial breed for a response a maximum genetic potential of all breed.

- Thai native chickens and their nutrient requirement was varied therefore a feed formulation to meet a precise requirement still not clear. The self-produce diet may undernutrition for all studied breeds especially in the case of low-utilizing feed ingredients using and we assumed that feed-utilization of studied breeds might different. Moreover, a commercial diet for Thai native chickens in Thailand is mainly to produce for cockfighting sport, not for meat production.

- We simulate a rearing condition of Thai native crossbred chicken production in Thailand which, almost producer rearing in an open system and applied a commercial diet for native chicken and its crossbred.

Revised text line: We add a reason on text in line 85-89 and 215-217 for additional description.

  1. Line 47. Commercial breeds of Gallus gallus also evolved from red jungle fowl. Why do you draw a distinction? Perhaps use the word selectively bred instead of ‘evolved’.

Response: We appreciate to change word following your comment.

Revised text line: line 46

  1. It is not clear in the abstract that KM1 and KM2 are cross breeds with the broiler breed. It also should be apparent that the authors are interested in meat production, not egg production. Usually, the purpose of crossbreeding programmes are to produce improved dual purpose chickens. Using a broiler chicken is a strange choice for a hybrid breeding expt.

Response:  We would like to describe a background of KM1 and KM2 breed. According to your mention that the purpose of crossbreeding programs are to produce improved dual purpose chickens, Research and development network center for animal breeding (Native Chicken) (NCAB), Khon Kaen University have researched on growth performance and meat tenderness of Thai native chicken. In 2006–2007, selection and testing of the traits commanded to four synthetic chicken breeds (Chee). The synthetic breeds were a cross bred chicken which established from Thai indigenous sire and commercial dam (Tanaosri). An initial, the study observed a variation of feather color patterns in crossbreds. Subsequently, the interesting feather color patterns are assorted including white, white with black hackle, black with white hackle and brown with yellowish bar hackle feather, and then curry color pattern is mated within group (Inter-se mating). Development of breed has been conducted for qualitative traits at least 5 generations. In 2014, four synthetic chicken breeds were named Thai chicken Kaen Tong KKU50 (brown with yellowish bar hackle feather), Thai chicken Soinin KKU50 (white color feather on body with black hackle), Thai chicken Soipet KKU50 (black color feather on body with white hackle) and Thai chicken Khai mook Isarn KKU50 (all white color feather) (Fig.1). However, this four-line of the synthetic breed has a different advantage so Khai mook Isarn KKU50 namely as KM has a property in meat production because of the highest growth rate among 4 breeds and moderate heat tolerance ability as well. while other lines focusing on a dual purpose and egg production. For this reason, KM was created as complementary breed to combine growth and heat tolerance merit and promoted as a meat production line and may upgrading the commercial breed percentage to 50 and 25% of Thai native chickens percentage of breed as KM1 and KM2. this study was to investigate the impact of this cross on fat deposition both and gene expression level.

Source: (Laopaiboon, 2015)

  1. Line 25 and throughout manuscript. Fraction is the incorrect word. Percentage is a more appropriate word.

Response: We appreciate to change word following your comment.

Revised text line: 24, 26

  1. Line 81 The Arbor Acre commercial broiler used in this study was purchased.

You need to make it clear that you have purchased hatching eggs. These will be from a four way cross so the genetics in the hybrids will be highly varied.

Response: In this study, we purchased as day-old chicks of commercial chickens in the breed of Arbor Acre from Charoen Pokphand Company and detailing of chickens obtaining and company who providing were added on text of material and method.

Revised text line: line 79-81

  1. Line 92 2.2. Slaughtering, fat deposition data and tissue collection

The authors do not mention that the broilers were slaughtered at 6 weeks of age.

Response: We describe detailing of slaughtering age of all breed in paragraph as “Twenty birds (10 males and 10 females) were randomly selected per genotype at 6, 8, 10 and 12 weeks of age for CH, KM1 and KM2. The commercial broiler which was slaughtered only at 6 weeks of age because of the limitations of raising broilers in open systems.”

Revised text line: line 94-96

References

1. Livak, K. J., and T. D. Schmittgen. 2001. Analysis of relative gene expression data using real-time quantitative PCR and the 2(-Delta Delta C(T)) Method. Methods 25:402–408.

2. Dridi, S., M. Taouis, A. Gertler, E. Decuypere, and J. Buyse. 2007. The regulation of stearoyl-CoA desaturase gene expression is tissue specific in chickens. J Endocrinol 192:229–236.

3. Wang, X. G., J. F. Yu, Y. Zhang, D. Q. Gong, and Z.-L. Gu. 2012. Identification and characterization of microRNA from chicken adipose tissue and skeletal muscle. Poultry science 91:139–49.

4. Laopaiboon, B. 2015. The Establishment of Four Synthetic Chicken Breeds.  Khon Kaen Agricultural. Journal. 43 (Supplement). 2.

Reviewer 2 Report

The manuscript entitled: “Increasing Fat Deposition via Upregulates the 2 Transcription of Peroxisome Proliferator-Activated 3 Receptor Gamma (PPARγ) in Native Crossbred 4 Chickens” described regulation of one of the most important axis in metabolism of fat tissue (regulation of fat content via PPARγ). Although this mechanism is commonly known in many animal species, in chickens it is relatively poorly described, which makes the manuscript interesting for the reader. The study is interesting, however I have a few questions/comments/suggestions for authors.

Major

  • Statistical analysis: Statistics is poorly described as well as it is not known why the authors adopted only one significance level (p <0.05) and not as is also assumed (p <0.01)
  • Description of statistics in figures. The use of letter markings misleads the reader because the authors refer to the graphs marked with letters in the figure legend.
  • Please provide more details on lipid determination.
  • Please include reference gene on the figures.

All gens and protein symbols should be styled according to species:

Eg.  Humans, non-human primates and chicken: Full name: peroxisome proliferator–activated receptor γ

Gene symbol: PPARG (italic)

Protein symbol: PPARγ

Mice and rats: Full name: peroxisome proliferator–activated receptor γ

Gene symbol: Pparg (italic)

Protein symbol: PPARγ

Author Response

Revision list

Manuscript ID: animals-1028268 “Increasing Fat Deposition via Upregulates the Transcription of Peroxisome Proliferator-Activated Receptor Gamma (PPARγ) in Native Crossbred Chickens”

Dear Dr. Cristina Maria Fiat and Reviewer,

We appreciate your responsibility as an editor and assistance with this manuscript. The reviewers have provided several useful suggestions to our manuscript and we have revised to the best of our ability. Our revised manuscript has modifications in yellow highlighted text, please let me know if I have to provide additional information.

Reviewer #2

  1. Statistical analysis: Statistics is poorly described as well as it is not known why the authors adopted only one significance level (p <0.05) and not as is also assumed (p <0.01)

Response: We appreciate your valuable comments to improve the quality of the manuscript. The statistical description for significance level both significantly different (p<0.05) and highly significantly different (p<0.01) were adding for symbol * and ** in all figures (1 to 3) for describe the level of significantly different in each analyzed week.

Revised text line:  Figure 1. (line 147-151), Figure 2. (line 187-189) and Figure 3 (line 191-193)

  1. Description of statistics in figures. The use of letter markings misleads the reader because the authors refer to the graphs marked with letters in the figure legend.

Response: We would like to thank you for your carefully check for the point that may mislead, the numbering (I, II, and III) was used for figure title legend instead of a, b and c. while the letter marking still used for the statistical marks. 

Revised text line: Figure 1. and legend (line 147-151), Figure 2. and legend (line 187-189) and Figure 3. and legend (line 191-193)

  1. Please provide more details on lipid determination.

Response: The detail of lipid determination was added in the text along with reference in material and method.

Revised text line: line 103-104 in material and method and line in references

  1. Please include reference gene on the figures.

Response: The reference gene (18S rRNA) was added in the figure legend to represent the gene for normalized gene expression of PPARA and PPARG in both figures 2 and 3.

Revised text line: Figure 2. and legend (line 187-189) and Figure 3. and legend (line 191-193)

  1. All genes and protein symbols should be styled according to species:

Response: According to your kindly comment, PPAR gamma and PPAR alpha genes in chickens (Gallus gallus) was assigned symbol by NCBI as PPARG and PPARA respectively

(source: https://www.ncbi.nlm.nih.gov/gene/374120 for PPARA and https://www.ncbi.nlm.nih.gov/gene/373928 for PPARG)

for this reason, we appreciate changing the symbol of these genes from PPARγ to PPARG and PPAR α to PPARA throughout this study.

Revised text line: throughout all text and figures of this study.

Round 2

Reviewer 2 Report

The authors adequately answered the questions and revised the manuscript.